UHPLC-ESI-QE-Orbitrap-MS based metabolomics reveals the antioxidant mechanism of icaritin on mice with cerebral ischemic reperfusion

Tang Yunfeng 1
Sun Lixin 2
Zhao Yun 1
Yao Jingchun 1
Feng Zhong 1 3
Liu Zhong 1
Zhang Guimin lunanzhangguimin@yeah.net 1
Sun Chenghong sch658@163.com 1
1 State Key Laboratory of Generic Manufacture Technology of Chinese Traditional Medicine, Lunan Pharmaceutical Group Co. Ltd. , Linyi , Shandong Province , China
2 Linyi Traditional Chinese Medicine Hospital , Linyi , Shandong Province , China
3 School of Pharmaceutical Sciences (Shenzhen), Sun Yat-sen University , Shenzhen , Guangdong Province , China
Gould Gwyn
Electronic publication date: 2023 Jan 10
Publication date: 2023
Volume: 11
Electronic Location ID: e14483
Received 2022 Sep 29; Accepted 2022 Nov 8
Copyright: ©2023 Tang et al.
Copyright year: 2023
Copyright holder: Tang et al.
License: This is an open access article distributed under the terms of the Creative Commons Attribution License, which permits unrestricted use, distribution, reproduction and adaptation in any medium and for any purpose provided that it is properly attributed. For attribution, the original author(s), title, publication source (PeerJ) and either DOI or URL of the article must be cited.
License URL: https://creativecommons.org/licenses/by/4.0/

Keywords: Cerebral ischemic reperfusion, Icaritin, Metabolomics, UHPLC-ESI-QE-Orbitrap-MS, Oxidative stress

Funding: Key Research and Develop Program (Major Scientific and Technological Innovation Project) of Shandong Province (No. 2021CXGC010508) This work was supported by the the Key Research and Develop Program (Major Scientific and Technological Innovation Project) of Shandong Province (No. 2021CXGC010508). The funders had no role in study design, data collection and analysis, decision to publish, or preparation of the manuscript.

==============================
Background

Icaritin (ICT) has been previously demonstrated to display protective effects against cerebral ischemic reperfusion (I/R) by inhibiting oxidative stress, but the mechanism remains unclear. This study aimed to explore the mechanism from the perspective of metabolomics.

Methods

A mice cerebral artery occlusion/reperfusion (MCAO/R) model was explored to mimic cerebral ischemic reperfusion and protective effect of ICT was assessed by neurologic deficit scoring, infarct volume and brain water content. Ultra-high-performance liquid chromatography electrospray ionization orbitrap tandem mass spectrometry (UHPLC-ESI-QE-Orbitrap-MS) based metabolomic was performed to explore potential biomarkers. Brain tissue metabolic profiles were analyzed and metabolic biomarkers were identified through multivariate data analysis. The protein levels of Nrf2, HO-1 and HQO1 were assayed by western blot. The release of malondialdehyde (MDA) and the activity of superoxide dismutase (SOD), glutathione peroxidase (GSH-Px) and catalase (CAT) were detected using corresponding assay kits.

Results

The results showed that after ICT treatment, the neurological deficit, cerebral infarction area, brain edema and the level of MDA in brain tissue of MCAO/R mice were significantly reduced. Meanwhile, ICT enhanced the activity of SOD, CAT and GSH-Px. Western blot results confirmed that ICT up-regulated the protein levels of antioxidant-related protein including Nrf2, HO-1 and NQO1. According to the metabolomic profiling of brain tissues, clear separations were observed among the Sham, Model and ICT groups. A total of 44 biomarkers were identified, and the identified biomarkers were mainly related to linoleic acid metabolism, arachidonic acid metabolism, alanine, aspartate and glutamate metabolism, arginine biosynthesis, arginine and proline metabolism, D-glutamine and D-glutamate metabolism, taurine and hypotaurine metabolism and purine metabolism, respectively. At the same time, the inhibitory effect of ICT on arachidonic acid and linoleic acid in brain tissue, as well as the promoting effect on taurine, GABA, NAAG, may be the key factors for the anti-neurooxidative function of mice after MCAO/R injury.

Conclusion

Our results demonstrate that ICT has benefits for MCAO/R injury, which are partially related to the suppression of oxidative stress via stimulating the Nrf2 signaling and regulating the production of arachidonic acid, linoleic acid, taurine, GABA, NAAG in brain tissue.

Introduction

According to the statistic report of the World Health Organization (WHO), stroke is the second leading cause of mortality in the world (Wu et al., 2021a; Wu et al., 2021b), of which ischemic stroke accounts for approximately 85% (Strong, Mathers & Bonita, 2007). Ischemic stroke is characterized by cerebral infarction in the area of stenosis or occlusion of arterial perfusion, leading to neurological dysfunction (Sacco et al., 2013). Aging, hypertension, diabetes, and hyperlipidemia can cause inflammation and promote cerebral atherosclerosis, eventually leading to cerebral artery occlusion and corresponding brain parenchymal damage (Miao et al., 2021). Ischemic stroke involves a complicated cycle of cellular and molecular mechanisms that are interrelated. The mechanisms include loss of cell ion homeostasis, free radical-mediated toxicity, energy failure, acidosis, generation of arachidonic acid products, increased intracellular calcium levels, cytokine-mediated cytotoxicity, and infiltration of leukocytes complement activation, disruption of the blood–brain barrier (BBB), excitotoxicity, and activation of glial cells (Jin & Leng, 2022; Woodruff et al., 2011).

Oxidative stress is closely related to the development of brain stroke (Allen & Bayraktutan, 2009). Due to the high demand for oxygen and the existence of large amounts of unsaturated lipids, the brain is exceedingly vulnerable to oxidative stress (Cutler et al., 2004). Reactive oxygen species (ROS), the most important oxygen free radical, is closely related to the pathology and development of ischemic cerebrovascular disease (Deshmukh et al., 2017). Normal brain tissue has a balance between cellular oxidative and antioxidant defense systems. Abnormally generated ROS disrupts this balance, induces lipid peroxidation, DNA fragmentation, cellular dysfunction, and apoptosis (Finkel & Holbrook, 2000; Ren, Hu, Zhou, 2018). Transcription factor nuclear factor erythroid 2-related factor 2 (Nrf2), a transcription factor that regulates the expression of various antioxidant enzymes, plays a key role in antioxidative stress to protect cells from apoptosis (Pajares, Cuadrado & Rojo, 2017; Suzuki & Yamamoto, 2015; Wardyn, Ponsford & Sanderson, 2015; Xu et al., 2021). In normal condition, kelch-like ECH-associated protein 1 (Keap1) associates with Nrf2 (Pallesen, Tran, Bach, 2018). Once activated, Nrf2 dissociates from keap1, translocates to the nucleus (Liu et al., 2018), and induces the generation of many antioxidant proteins, including superoxide dismutase (SOD), glutathione peroxidase (GSH-Px), catalase (CAT), heme oxygenase-1 (HO-1), and NAD(P) H-quinone dehydrogenase 1 (NQO-1) (Uruno & Motohashi, 2011), to resist oxidative stress damage.

Cerebral ischemic reperfusion injury is an incurable disease with a high recurrence rate. Although the use of antiplatelet drugs and statins in primary and secondary stroke prevention is widespread, new and effective strategies are still needed. Natural medicine is an important source of modern medicine development (Liu et al., 2018). Icaritin (ICT, CAS#: 118525-40-9) is an in vivo metabolite of icariin (Hwang et al., 2018) isolated from the traditional Chinese medicine Epimedium sagittatum maxim (Ye, Yu & Zhao, 2020). It has a series of biological activities, including neuroprotection (Xu et al., 2021), immunomodulation (Lai et al., 2013) and anticancer effects (Huang, Zhu & Lou, 2007; Guo et al., 2011; Tong et al., 2011), and inducing the proliferation and differentiation of hematopoietic stem cells (Sun et al., 2018). ICT can improve the memory and learning ability of experimental Alzheimer’s disease mice (Li et al., 2020), and inhibit neuroinflammation and oxidative stress in Parkinson’s disease mice (Wu et al., 2021a; Wu et al., 2021b). Recently, our research confirmed that ICT had a significant protective effect on focal cerebral ischemia reperfusion injury in mice (Sun et al., 2017).

Metabolic disorder is a key event leading to cerebral ischemic reperfusion. The emergence of metabolomics analysis technology provides an effective tool for identifying these key metabolic biomarkers with potential diagnostic and prognostic value after ischemic stroke (Au, 2018). Metabolomics developed rapidly in the mid-1990s and is based on the qualitative and quantitative analysis of the final products in specific organisms or cells (Nicholson et al., 2002). As a bridge between genotype and phenotype, metabolomics can determine the overall changes in diseases by analyzing large databases, and clarify specific mechanisms from a systematic perspective. Compared with isolated single pathways or single biomarkers, systematic data is more conducive to elucidating the pathogenesis of complex diseases such as cerebral ischemic reperfusion (Griffin et al., 2011).

Although the antioxidant effect of ICT on middle cerebral artery occlusion/reperfusion (MCAO/R) injury has been reported earlier, its in-depth pharmacological mechanism is still unclear. In this study, we established a MCAO/R mice model and use metabolomics methods to study the potential mechanism.

Material and Methods

Animals

ICR mice (30∼34 g, 6∼7 weeks) were purchased from Beijing Vital River Laboratory Animal Technology Co., Ltd. (Beijing, China) with experimental animal use license SYXK (Lu) 2018 0008. All mice were housed in the specific pathogen free (SPF) feeding system under 12 h/12 h light and dark cycle under a standardized temperature at 20–26 °C and relative humidity at 40%–70% with 5 mice per cage, and were free access to water. All experimental animals were performed in strict compliance with the National Institutes of Health Guide for the Care and Use of Laboratory Animals. Procedures were approved by the Ethics Committee for Experimental Animals at State Key Laboratory of Generic Manufacture Technology of Chinese Traditional Medicine (approved on July 3rd, 2020; No. NH-IACUC-2020–047) for minimizing animal suffering. Mice in each cage are distinguished by different colored and marked labels.

Mice MCAO/R model

The mice MCAO/R model was established according to the method provided by Song et al. (2012). The mice were anesthetized with sodium pentobarbital (50 mg/kg, i.p.), and the right common carotid artery was exposed to the bifurcation of external and internal carotid artery (ECA and ICA). Insert a monofilament nylon suture (φ 0.20  ± 0.01 mm) into the ECA from the bifurcation of the common carotid artery for 8–10 mm into the ICA until a slight resistance was felt. Afterwards, the skin incision was sutured, and 1 h after MCAO, reperfusion was allowed by withdrawal of the suture thread until the tip cleared the ICA. Sham-operated mice were exposed to the same experimental procedure without occlusion of ECA. Animals were closely monitored with body temperature kept at (37  ± 0.5) °C.

Groups and drug administration

66 Male ICR mice (n = 22 for each group) were randomly divided into Sham group, Model group and ICT group according to body weight. The ICT group was pre-treated with ICT nanocrystalline (12 mg/kg/d, i.g.) (production batch No.: 190602, Lunan Pharmaceutical Group Co. Ttd., Linyi, Shandong, China) once a day for 3 days, and the normal group and model group received equal volume of deionized water. 10 min after the last dose, the Model group and ICT group received MCAO/R. A total of 24 h after reperfusion, all of the mice were anesthetized with sodium pentobarbital (50 mg/kg, i.p.) and sacrificed by bloodletting, and brain tissues were taken for corresponding tests. At the end of the experiment, the corpses were temporary stored in freezer at minus 20 °C.

Neurological deficits

The neurological deficit scores were evaluated with a method developed by Longa et al. (1989). The scores were as follows: 0, no neurological deficit; 1, failure to fully extend the contralateral forelimbs; 2, contralateral circling; 3. leaning to the contralateral side; 4. unconsciousness.

Brain water content

The wet-dry weight method was used to evaluate the brain edema (Paczynski et al., 1997). The brains (n = 6 for each group) were rapidly removed, the water and blood on the surface were quickly cleaned. Brain tissues were weighed and recorded as wet weight. Then, the fresh brains were dried in a 70 °C oven to acquire the dry weight. Brain water content was calculated with the formula as follows: Brain water content (%) = (wet weight − dry weight)/(wet weight) × 100%.

Cerebral infarction detection

The brains (n = 6 for each group) were quickly removed and cut into five coronal slices of one mm thickness, immersed in PBS solution containing 1.5% 2,3,5-triphenyltetrazolium chloride (TTC) (Sigma-Aldrich, St. Louis, MO, USA) and incubated at 37 °C for 30 min. Afterwards, the stained slides were photographed with a digital camera and analyzed by ImageJ analysis software (National Institutes of Health, Bethesda, MD, USA).

Brain sample collection and preparation for metabolomics study

The injury side brain tissue (n = 10 for each group) was taken out and homogenized in pre-cooled 80% methanol (1.5 mL/50 mg) for 1 min, vortex for 2 min, and centrifuge at 12,000 rpm at 4 °C for 15 min. The supernatant was collected, and another 1.5 mL of pre-cooled CH2Cl2/MeOH (3:1, v/v) was used to extract the precipitate again. The supernatants of the two extractions were mixed and dried by nitrogen, and then reconstituted by 300 µL methanol. The sample was centrifuged at 12,000 rpm (4 °C) for 10 min, and 2 µL of the supernatant was used for analysis.

Ultra-high-performance liquid chromatography electrospray ionization orbitrap tandem mass spectrometry analysis

An ultra-high-performance liquid chromatography electrospray ionization orbitrap tandem mass spectrometry (UHPLC-ESI-QE-Orbitrap-MS) (Thermo Fisher Scientific, Inc., Waltham, MA, USA) technology was used to characterize the differences in tissue metabolism to screen out the potential different metabolites and possible metabolic pathways in brain according to the method of Lu et al. (2022) that meet the sample requirements. Metabolites were separated on a Hypersll GOLD C18 column (Hypersil GOLD 100 × 2.1 mm, 3.0 µm) (Thermo Fisher Scientific, Inc., Waltham, MA, USA) at a column temperature of 40 °C. The mobile phase was consisted with 0.1% formic acid (A) and methanol (B). The flow rate was set to 0.2 mL/min and the injection volume was 2 µL. The auto-sample temperature was 10 °C. The specific elution condition is in Table 1.

An electrospray ionization source (ESI) was used in both positive and negative ion mode, and mass spectrometric parameters were as follows: the scanning range was 50–1,075 m/z, spray voltage, ion source temperature 350 °C; sheath gas 40 kPa, auxiliary gas 10 kPa. The spectrum was acquired in the orbitrap mode, and the induced collision dissociation energy was 20, 40 and 60, respectively. The resolution of MS1 was 70,000, and the detector voltage was 4.0 kV. The MS2 spectrum was acquired in the orbitrap mode, the scanning range was 50–1,075 m/z; the resolution of MS2 was 17,500, and the detector voltage was 2.8–4.0 kV.

All data were acquired and processed using Compound Discover 3.1 software (Thermo Fisher Scientific, Inc., Waltham, MA, USA) to obtain matching and aligned peak data. The setting parameters were as follows: mass range was 50–1,075, mass deviation was 5 ppm, the retention time deviation was 0.05 min, and the signal-noise ratio threshold was 3. The resultant data matrices were imported to SIMCA-P 14.1. The data of principal component analysis (PCA) and orthogonal partial least squares discriminant analysis (OPLS-DA) were obtained. Additionally, permutation tests should be presented for external validation to evaluate the fitting of the model visually. The biomarkers were screened and confirmed with the restriction of variable importance (VIP) value (VIP >1) and t-test (P < 0.05) in projection. Finally, the biological interpretation and related metabolic pathways of the identified differential metabolites were depended on KEGG (Kyoto Encyclopedia of Genes and Genomes) database (http://www.genome.jp/kegg/) and MetaboAnalyst 5.0 (http://www.metaboanalyst.ca/).

Table 1 Gradient elution condition of mobile phase.

t/min	A/%	B/%	
0	95	5	
1.0	95	5	
9.0	5	95	
12.0	5	95	
12.1	95	5	
15.0	95	5	

MDA, SOD, GSH-Px and CAT assay

Levels of MDA (Cat#: A003-1-2) and the activity of SOD (Cat#: A001-1-2), GSH-Px (Cat#: A005-1-2) and CAT (Cat#: A007-1-1) (Nanjing Jiancheng Bioengineering Institute, Nanjing, Jiangsu, China) in brain tissue were evaluated by respective kits according to the manufacturer’s instructions. MDA, SOD, GSH-Px and CAT were measured spectrophotometrically at a wavelength of 532 nm, 450 nm, 405 nm and 405 nm using a microplate spectrophotometer (Thermo Fisher Scientific, Inc., Waltham, MA, USA).

Western blot

From the samples used in metabonomics study, three remaining samples were randomly selected in each group for western blot study. Total protein extracted from dorsal skin tissue was treated with RIPA lysis buffer (Beyotime Institute of Biotechnology, Shanghai, China) and centrifuged at 12,000 rpm for 5 min at 4 °C. The BCA assay kit (Beyotime Institute of Biotechnology, Shanghai, China) was used to explore the protein concentration in the supernatant. Equal amount of protein samples was separated on sodium dodecyl sulfate polyacrylamide gel electrophoresis (SDS-PAGE) gels, and transferred to PVDF membranes (Millipore, Billerica, MA, USA). After blocked with 5% BSA (Beyotime Institute of Biotechnology, Shanghai, China) for 2 h, the membranes were incubated with primary antibody against Nrf2 (1:1,000, Cell Signaling Technology, Danvers, MA, USA), HO-1 (1:1,000, Cell Signaling Technology, Danvers, MA, USA), NQO-1 (1:1,000, Cell Signaling Technology, Danvers, MA, USA) and α-tubulin (1:1,000, Beyotime Institute of Biotechnology, Shanghai, China) at 4 °C overnight. The samples were incubated with horseradish peroxidase secondary antibody (Beyotime Institute of Biotechnology, Shanghai, China). The protein bands were visualized using enhanced chemiluminescence (ECL) (Beyotime Institute of Biotechnology, Shanghai, China) method. Band density was calculated with a ChemiScope 6200 Chemiluminescence imaging system (Clinx Science Instruments Co., Ltd., Shanghai, China).

Statistical analysis

All data were statistically analyzed using SPSS19.0 and expressed as mean ± standard deviation (SD). Statistical analysis was performed using a one-way analysis of variance (ANOVA). Independent-sample t test was used to compare differences between two groups. P < 0.05 was considered of statistical significance.

Results

Neuroprotective effects of ICT on MCAO/R mice

Initially, we investigated and confirmed the protective effects of ICT against MCAO/R in ICR mice. As shown in Fig. 1, 24 h after reperfusion, ratios of infarct volume (Fig. 1A), brain water contents (Fig. 1B) and neurological deficit scores (Fig. 1C) in MCAO/R mice were significantly higher than mice of Sham group. Because the Sham group brains had no brain tissue damage, the brain slices were completely stained in red and counted the infarct volume as zero (Fig. 1A). MCAO/R mice pre-treated (i.g.) with ICT at dosage of 12 mg/kg significantly decreased the brain infract volume (Fig. 1A), inhibited MCAO/R-induced brain edema (Fig. 1B), and diminished the neurological deficit scores (Fig. 1C), which compared to the MCAO/R group.

Figure 1 Neuroprotective effects of ICT on MCAO/R mice.

(A) Representative TTC-stained coronal sections of each group (a) and the statistical results (b). The infarct area was unstained, and the normal part was stained in red. (B) Brain water content. (C) The neurological deficit scores result. Data are mean ± SD, n = 6. **P < 0.01 vs Sham group, ##P < 0.01 vs Model group.

Metabolic profiling of brain tissue

The representative UHPLC-ESI-QE-Orbitrap-MS total ion chromatogram of the brain tissue samples in each group is shown in Fig. 2. The peak shape of each substance is good and the peaks are well separated from each other, indicating that the chromatographic and MS conditions were suitable for the measurement of the samples in this study.

Figure 2 The typical base peak chromatograms of QC, Sham, Model and ICT samples in the positive mode (A) and the negative mode (B), respectively.

A total of 1,318 peaks were merged and imported into the SIMCA-P 14.1 software for multivariate statistical analysis. As the commonly used unsupervised method in metabolomics studies, the PCA model was applied to show the intrinsic separation among the Sham, Model and ICT groups. The scores plot of PCA (Fig. 3A) showed that the Sham and Model groups were clearly distinct, with the ICT group being much closer to the Sham group than the Model group. This observation demonstrated that ICT could alleviate the MCAO/R-induced metabolic abnormalities.

Figure 3 PCA and OPLS-DA based on UHPLC-ESI-QE-Orbitrap-MS data from brain tissue of all groups.

(A) PCA scores plot of Sham group, Model group and ICT group. (B) OPLS-DA score plot of the MCAO/R group and the Sham group. (C) OPLS-DA score plot of the ICT group and the Model group. (D) Permutation test of Model group and Sham group. (E) Permutation test of ICT group and Model group.

In order to eliminate any non-specific effects and confirm the biomarkers, OPLS-DA was used to analyze the metabolic profiles of the Sham, Model and ICT groups of samples. As shown in the OPLS-DA score plot (Figs. 3B and 3C), a clear separation between Sham and Model groups, as well as Model and ICT groups were observed. The cumulative R2X, R2Y and Q2 were 0.885, 0.998, 0.747 (Sham group vs Model group) and 0.853, 0.995, 0.878 (Model group vs ICT group) in the OPLS-DA model, respectively. According to the results of chance permutation, no overfitting was observed (Figs. 3D and 3E). In addition, all the green R2 values on the left were lower than the original points on the right, which implies that the original model is feasible.

Identification of biomarkers

In this experiment, VIP > 1 in the OPLS-DA model, with P < 0.05 were determined to be the screening criteria for differential metabolites. The chemical structures of important metabolites were then determined by searching online database HMDB (human metabolome database) (http://www.hmdb.ca/) using the data of m/z obtained by UPLC-ESI-QE-Orbitrap-MS with high resolution. Subsequently, a total of 44 potential biomarkers, such as L-aspartic acid, L-glutamic acid, guanine, gamma-aminobutyric acid (GABA), N-acetylaspartylglutamic acid (NAAG), 5-aminoimidazole ribonucleotide, guanosine monophosphate (GMP), adenine, adenosine, inosine, uric acid, taurine, arachidonic acid, linoleic acid, and so on of MCAO/R mice (Table 2) were identified. After oral administration of ICT, all the 44 identified biomarkers fell in the levels between the Sham and the Model groups (Table 2). The heat map revealed their relative changes in Sham, Model and ICT groups (Fig. 4).

Table 2 Potential biomarkers of MCAO/R injury pre-treatment with ICT.

No.	Name	Formula	M/Z	RT (min)	Ion mode	HMDB	Model vs Sham	ICT vs Model	
1	LysoPC(18:2)	C26H50NO7P	519.33119	11.706	ESI+	HMDB0010386	↑***	↓****	
2	LysoPC(20:4(5Z,8Z,11Z,14Z)/0:0)	C28H50NO7P	543.33117	11.643	ESI+	HMDB0010395	↑***	↓****	
3	Gamma-Aminobutyric acid (GABA)	C4H9NO2	103.0636	1.375	ESI+	HMDB0000112	↓***	↑******	
4	L-Pyroglutamic acid	C5H7NO3	129.04231	1.68	ESI+	HMDB0000267	↓***	↑*****	
5	5-Aminoimidazole ribonucleotide	C8H14N3O7P	295.05808	1.346	ESI+	HMDB0001235	↓***	↑*****	
6	2-Aminoacrylate	C3H5NO2	87.03239	1.814	ESI+	HMDB0003609	↓***	↑*****	
7	DL-Carnitine	C7H15NO3	161.10482	1.376	ESI+	HMDB0000062	↓***	↑*****	
8	2-Hydroxyethanesulfonate	CH5O4P	111.99258	1.522	ESI+	HMDB0003903	↓***	↑*****	
9	Piperidine	C5H11N	85.08955	3.026	ESI+	HMDB0034301	↓	↑****	
10	gamma-Glutamylglutamic acid	C10H16N2O7	276.09512	1.658	ESI+	HMDB0011737	↓***	↑*****	
11	Valine	C5H11NO2	117.07896	1.698	ESI+	HMDB0000883	→	↓*****	
12	Docosahexaenoic acid	C22H32O2	328.23984	12.149	ESI−	HMDB0002183	↑***	↓*****	
13	Stearic acid	C18H36O2	284.27113	13.556	ESI−	HMDB0000827	↑***	↓****	
14	Palmitic acid	C16H32O2	256.23975	12.656	ESI−	HMDB0000220	↑***	↓*****	
15	Arachidonic acid	C20H32O2	304.23981	12.227	ESI−	HMDB0001043	↑*	↓*****	
16	Linoleic acid	C18H32O2	280.23987	12.323	ESI−	HMDB0000673	↑***	↓******	
17	Sphingosine	C18H37NO2	299.28203	11.981	ESI−	HMDB0000252	↑***	↓*****	
18	8Z,11Z,14Z-Eicosatrienoic acid	C20H34O2	306.25557	12.567	ESI−	HMDB0002925	↑***	↓*****	
19	Uric acid	C5H4N4O3	168.0273	1.729	ESI−	HMDB0000289	↑***	↓****	
20	2-C-Methyl-D-erythritol 4-phosphate	C5H13O7P	216.03938	1.335	ESI−	METPA1027	↑**	↓****	
21	Citric acid	C6H8O7	192.02605	2.172	ESI−	HMDB0000094	↑***	↓****	
22	Margaric acid	C17H34O2	270.25567	13.084	ESI−	HMDB0002259	↑**	↓****	
23	LysoPE(18:1(9Z)/0:0)	C23H46NO7P	479.30089	11.956	ESI−	HMDB0011506	↑***	↓*****	
24	LysoPE(18:0/0:0)	C23H48NO7P	481.31588	13.529	ESI−	HMDB0011130	↑**	↓****	
25	LysoPE(20:4(5Z,8Z,11Z,14Z)/0:0)	C25H44NO7P	501.28537	11.571	ESI−	HMDB0011517	↑**	↓****	
26	LysoPE(22:6(4Z,7Z,10Z,13Z,16Z,19Z)/0:0)	C27H44NO7P	525.28508	11.53	ESI−	HMDB0011526	↑**	↓****	
27	N-Acetyl-L-aspartic acid	C6H9 NO5	175.04712	1.59	ESI−	HMDB0000812	↓***	↑****	
28	4-Oxoproline	C5H7NO3	129.04136	1.266	ESI−	METPA0228	↓***	↑****	
29	Adenosine	C10H13N5O4	267.09651	4.436	ESI−	HMDB0000050	↓***	↑****	
30	Taurine	C10H13N5O4	267.09651	4.436	ESI−	HMDB0000251	↓***	↑****	
31	L-(+)-Lactic acid	C3H6O3	90.03027	1.631	ESI−	HMDB0000190	↓*	↑****	
32	Inosine	C10H12N4O5	268.08053	2.599	ESI−	HMDB0000195	↓***	↑****	
33	Adenine	C5H5N5	135.05324	4.438	ESI−	HMDB0000034	↓***	↑****	
34	Uridine	C9H12N2O6	244.06909	2.032	ESI−	HMDB0000296	↓***	↑****	
35	N-Acetylaspartylglutamic acid (NAAG)	C11H16N2O8	304.09041	1.253	ESI−	HMDB0001067	↓***	↑****	
36	N-Acetylneuraminic acid	C11H19NO9	309.10667	1.468	ESI−	HMDB0000230	↓*	↑****	
37	1-Pyrroline-4-hydroxy-2-carboxylate	C5H7NO3	129.04248	1.262	ESI−	HMDB0002234	↓***	↑****	
38	L-Aspartic acid	C4H7NO4	133.03624	1.265	ESI−	HMDB0000191	↓***	↑*****	
39	L-Glutamic acid	C5H9NO4	147.05195	1.27	ESI−	HMDB0000148	↓***	↑****	
40	Oxidized glutathione	C20H32N6O12S2	612.15137	1.254	ESI−	HMDB0003337	↓***	↑****	
41	Guanine	C4H4N6O	152.04411	1.859	ESI−	HMDB0000132	↓***	↑****	
42	N-Acetyl-L-alanine	C5H9NO3	131.05697	1.629	ESI−	HMDB0000766	↓***	↑****	
43	Guanosine monophosphate (GMP)	C10H14N5O8P	363.05775	1.461	ESI−	HMDB0001397	↓**	↑****	
44	UDP-alpha-D-galactose	C15H24N2O17P2	566.05448	1.704	ESI−	HMDB0000302	↓**	↑****	
Notes.

* P < 0.05.

** P < 0.01.

*** P < 0.001 vs Sham group.

**** P < 0.05.

***** P < 0.01.

****** P < 0.001 vs Model group.

Metabolic pathway analysis

All of the 44 biomarkers were imported to MetaboAnalyst 5.0 (McGill University, Montreal, QC, Canada) to analyze metabolic pathways. The criterion of the potential targeted pathway was set at impact >0.1 in MetaboAnalyst computation. The results show that ICT protects cerebral MCAO/R injury by regulating several metabolic pathways, including linoleic acid metabolism, arachidonic acid metabolism, alanine, aspartate and glutamate metabolism, arginine biosynthesis, arginine and proline metabolism, D-glutamine and D-glutamate metabolism, taurine and hypotaurine metabolism and purine metabolism (Fig. 5).

Effect of ICT on antioxidant activity in mice brain with MCAO/R

To further investigate the effect of ICT on oxidative stress in the brain tissue of MCAO/R mice, we detected the protein levels of Nrf2, HO-1 and NQO-1 by western blot. The results showed that, compared with the Sham group, the protein levels of Nrf2 (Fig. 6Aa), HO-1 (Fig. 6Ab) and NQO-1 (Fig. 6Ac) in the brain tissue of the mice in the model group were significantly decreased. Treatment with ICT upregulated the levels of these proteins in the ICT group compared to the Model group (Fig. 6A).

Figure 4 Heat map of metabolites in the cerebral ischemic reperfusion injury of rats from Sham, Model and ICT groups.

Figure 5 Pathway analysis of the differential metabolites reversed by ICT.

The color and size of each circle represented the P value and the pathway impact factor, respectively. MetaboAnalyst 5.0 was applied to achieve this function. A, Linoleic acid metabolism; B, Alanine, aspartate and glutamate metabolism; C, Purine metabolism; D, Arginine biosynthesis; E, Arginine and proline metabolism; F, D-Glutamine and D-glutamate metabolism; G, Taurine and hypotaurine metabolism; H, Arachidonic acid metabolism.

Figure 6 ICT protects against oxidative stress in brain tissue of MCAO/R mice.

(A) Representative images and quantitative analysis of Nrf2 (a), HO-1 (b) and NQO-1 (c) in the brain tissue. Column diagrams for MDA (A), SOD (B), GSH-Px (C) and CAT (D). Results were the mean ± SD, n = 3 for A and n = 10 for B, C, D, and E. ** P < 0.01 vs NC group; # P < 0.05 and ## P < 0.01 vs Model group.

Furthermore, we detected the levels of oxidative stress-related downstream MDA and the activities of SOD, GSH-Px and CAT, and found that the level of MDA (Fig. 5B) in the brain tissue of the model group mice was significantly increased, while the activities of SOD (Fig. 6C), GSH-Px (Fig. 6D) and CAT (Fig. 6E) were significantly decreased. ICT treatment significantly weakened the effect of MCAO/R on MDA (Fig. 6B) and increased SOD (Fig. 6C), GSH-Px (Fig. 6D) and CAT (Fig. 6E) activity. These results clearly demonstrate that ICT protects mice from MCAO/R injury by inhibiting oxidative stress in brain tissue.

Discussion

Early studies confirmed that ICT can protect cerebral ischemia reperfusion injury by inhibiting neuroinflammation and oxidative stress (Sun et al., 2017). This study found that ICT inhibited MCAO/R-induced brain tissue necrosis and brain edema, as well as improved neurological scores in mice once again. At the same time, we found that ICT can inhibit oxidative stress in brain tissue by upregulating Nrf2 to activate the expression and activity of a series of antioxidant proteins. Furthermore, brain tissue metabolomic profiling of each group was performed by UHPLC-ESI-QE-Orbitrap-MS, and a total of 44 endogenous metabolites were identified. These endogenous metabolites participate in metabolic pathways such as polyunsaturated fatty acid (PUFA) metabolism (linoleic acid metabolism and arachidonic acid metabolism), amino acid metabolism (alanine, aspartate and glutamate metabolism, arginine biosynthesis, arginine and proline metabolism, D-glutamine and D-glutamate metabolism, taurine and hypotaurine metabolism) and purine metabolism.

PUFA metabolism

After cerebral ischemic reperfusion, a large number of free radicals are generated through multiple pathways such as arachidonic acid metabolism and linoleic acid metabolism, which then produces neurotoxic effects on intracellular proteins, lipids, nucleotides and other components, causing nerve cell damage (Abramov, Scorziello & Duchen, 2007). The neurotoxic effects of oxygen free radicals can be summarized as: (1) Acting on polyvalent unsaturated fatty acids, causing lipid peroxidation; (2) Inducing cross-linking of macromolecular substances such as DNA, RNA, polysaccharides, and amino acid, which leading to lose their original activity or function (Wang et al., 2013). In this experiment, we found that the levels of PUFA in brain tissues, such as linoleic acid and arachidonic acid (ARA), were significantly increased. ARA is an omega-6 polyunsaturated fatty acid produced from linoleic acid (Hoda et al., 2009; Hirasawa, 2019), and the levels of ARA and linoleic acid in the Model group were significantly higher than those in the Sham group, while the content of these PUFA in the ICT treated group was significantly lower than that in the Model group. Our earlier study indicated that ICT could reduce the level of brain peroxidation in MCAO/R mice (Sun et al., 2017), which may be due to the fact that ICT can reduce the content of ARA and linoleic acid in brain tissue.

Amino acid metabolism

The results of this study suggested that 5 endogenous metabolites, including taurine, L-glutamic acid, L-aspartic acid, NAAG and GABA, which participated in alanine, aspartate and glutamate metabolism, arginine biosynthesis, arginine and proline metabolism and D-glutamine and D-glutamate metabolism, were decreased following MCAO/R-induced cerebral ischemia.

Glutamic acid and aspartic acid are excitatory amino acids that can stimulate postsynaptic neurons, while taurine, GABA and NAAG are inhibitory amino acids that can reduce the excitability of neurotransmission. Glutamic acid and aspartic acid play important roles in the process of ischemic neuronal death (Choi & Rothman, 1990), and inhibition of excitatory amino acid release can reduce cerebral ischemic reperfusion injury (Graham et al., 1993). Preliminary studies have shown that the extracellular content of excitatory amino acids (glutamic acid and aspartic acid) and inhibitory amino acids (GABA, taurine) in rats reached the maximum 1–2 h after MCAO/R, but began to decrease 3 h after MCAO/R (Melani et al., 1999). The decrease in glutamic acid and aspartic acid levels may be due to insufficient energy production caused by neuropathy after MCAO/R. This study also found that after 24 h of ischemia-reperfusion, mice brain tissue glutamic acid and aspartic acid were significantly reduced, which is consistent with the above report.

Taurine is a sulfur-containing non-essential amino acid that does not directly participate in protein biosynthesis, but plays a key role in the metabolism of cystine and cysteine (Chen et al., 2019). In addition, it is reported that taurine has antioxidant and anti-inflammatory activities, which can inhibit oxidative stress by promoting glutathione (GSH) biosynthesis, regulate the production of reactive oxygen species (ROS) in mitochondria to combat oxidative stress (Hung, 2006), and inhibit neuroinflammation (Seol et al., 2021). Gharibani et al. (2015) found that taurine can reduce neurological deficits and cerebral infarction volume caused by MCAO/R. In this experiment, the NAAG of MCAO/R mice was down-regulated, and NAAG content was up-regulated after ICT treatment, indicating that NAAG is related to the protective effect of ICT on MCAO/R. After nerve injury, the excessive release of glutamate is often accompanied by the release of NAAG (Zhong et al., 2009; Zhong, Luo & Jiang, 2014). Over-released glutamic acid induces ROS production and inhibit GSH synthesis (Cao et al., 2016). NAAG can activate the type 3 metabotropic glutamate receptor (mGluR3) in brain tissue, promote glutamate uptake by astrocytes and inhibit glutamate release (Yourick et al., 2003; Zuo et al., 2012). However, the released NAAG is quickly hydrolyzed by glutamate carboxypeptidase II to N-acetylaspartic acid, thereby losing the neuroprotective effect (Yao et al., 2005; Zhong et al., 2006). GABA is another important inhibitory neurotransmitter, and its metabolic changes may be an important factor in neuronal damage caused by cerebral ischemia (Kang et al., 2002), GABA level rises sharply in the early stage of cerebral ischemia injury, causing post-synaptic neurons. Meanwhile, GABA can reduce the release of glutamate through pre-synaptic neurons inhibition, and assist in the transport of Ca2+ to maintain osmotic pressure inside and outside the cell (Olthof & Verhoef, 2005) to reduce cell damage. Our previous studies have found that ICT protects mice from MCAO/R damage is closely related to the inhibition of brain oxidative stress and neuroinflammation (Sun et al., 2017). In this experiment, the content of taurine, GABA and NAAG in the brain tissue of the ICT intervention group increased, and the area of cerebral infarction decreased. These could explain the mechanism why ICT protects brain tissue from oxidative stress and neuroinflammatory injury.

Purine metabolism

ICT significantly corrected the disorder of purine metabolism in the brain tissue of MCAO/R mice. Purine metabolism disorders may cause, or reflect, defective DNA synthesis or repair, and further cause neurons to undergo apoptosis (Pang et al., 2012; Wang, 2016). The production of ATP in necrotic cells is an important factor in inducing neurooxidative stress in ischemic cerebrovascular disease. In stroke patients, ATP can promote the release of ROS from mitochondria in brain tissue cells, thereby inducing oxidative stress (Chen et al., 2022). Moreover, previous studies have shown that under ischemic conditions, due to insufficient energy supply in brain tissue, adenosine triphosphate (ATP) is broken down into adenosine, further into inosine and xanthine (Barsotti & Ipata, 2004), and finally decomposed into uric acid (Su et al., 2016). Therefore, the consumption of inosine and adenosine may be related to the catabolism of ATP. Adenosine has always been considered as a potential immunomodulatory and neuroprotective agent (Sweeney, 1997; Cunha, 2001), and plays an important role in regulating ischemic neuronal injury. Adenosine can reduce the release of pre-synaptic neuronal excitatory amino acids, and inhibit post-synaptic neuronal membrane hyperpolarization and NMDA (N-methyl-D-aspartic acid receptor) receptor activation. These effects can limit the influx of Ca2+ into neurons and exert neuroprotective effects (Schubert et al., 1994; De Mendonca, Sebastiao & Ribeiro, 2000). Inosine has potent immunomodulatory and neuroprotective effects. It can inhibit the release of proinflammatory mediators produced by macrophages, and protect MCAO/R-induced nerve damage by reducing immune activation (Hasko, Sitkovsky & Szabo, 2004). After treatment with ICT, the level of 5-aminoimidazole ribonucleotide, GMP, guanine, adenosine, adenine and inosine showed a trend toward normalcy. These results suggest that the therapeutic effects of ICT on cerebral ischemic are partially due to interferences with purine metabolism.

In summary, ICT can protect mice from MCAO/R injury by regulating various metabolic pathways such as PUFA metabolism, amino acid metabolism and purine metabolism. The results of metabolomics have repeatedly suggested that the protection of MACO/R damage by ICT is closely related to the inhibition of oxidative stress. In particular, ICT inhibited the release of arachidonic acid and linoleic acid in model mice, and up-regulated the levels of taurine, GABA and NAAG, which is crucial for improving the ability of mice to resist MCAO/R-induced oxidative stress injury in brain tissue. At the same time, the production of oxidative stress will in turn inhibit the production of these metabolites. Both of them regulate each other and jointly maintain the balance of redox level to ensure the healthy state of brain tissue. To further verify the in vivo antioxidant effect of ICT, we detected the expression of oxidative stress-related proteins Nrf2, HO-1 and NQO-1 in mouse brain tissue. The results showed that after ICT treatment, the expression of the above proteins of the model mice were significantly increased. Meanwhile, the content of MDA in the brain tissue was further reduced, and the activities of SOD, GSH-Px and CAT were increased. These results further verified from the protein level that ICT can protect mice from MCAO/R injury by inhibiting oxidative stress.

Conclusion

After ICT treatment, the neurological deficit, cerebral infarction area and brain edema of MCAO/R mice were significantly reduced, indicating that ICT has a protective effect on MCAO/R injury. Our study also found that ICT can play a role in resisting neurooxidative stress by up-regulating the level of Nfr2. To further elucidate the mechanism by which ICT exerts its protective effect on the brain, we developed a metabolomics approach to explore the mechanism of ICT protection against MCAO/R injury. The results indicate that the protective effect of ICT on cerebral ischemia is partly attributable to PUFA metabolism, amino acid metabolism and purine metabolism. At the same time, the inhibitory effect of ICT on arachidonic acid and linoleic acid in brain tissue, as well as the promoting effect on taurine, GABA, NAAG, may be the key factors for the anti-neurooxidative function of mice after MCAO/R injury. In short, the activation of Nrf2 signal and the changes of brain tissue metabolites are mutually regulated, which jointly participate in the protective effect of ICT on MCAO/R mice. However, this study still did not clearly elaborate the relationship between the regulative effects of ICT on metabolites and inhibitive effect on oxidative stress, and a large number of experiments are still needed for further research. Meanwhile, there are still large discrepancies between the results of preclinical animal studies and real clinical applications. Therefore, further clinical studies are still needed to confirm the results of this study after the drug is on the market.

Supplemental Information

Supplemental Information 1 Original figures for cerebral infarction

Click here for additional data file.

Supplemental Information 2 Infarct volume, brain water content and neurological deficits

Click here for additional data file.

Supplemental Information 3 Original data for metabolomics

Click here for additional data file.

Supplemental Information 4 Original data for western blot

Click here for additional data file.

Supplemental Information 5 Infarct volume, brain water content and neurological deficits

Click here for additional data file.

Additional Information and Declarations

Competing Interests

Author Contributions

Animal Ethics

Data Availability

Yunfeng Tang, Yun Zhao, Jingchun Yao, Zhong Feng, Zhong Liu, Guimin Zhang and Chenghong Sun are employed by Lunan Pharmaceutical Group Co. Ltd.

Yunfeng Tang conceived and designed the experiments, performed the experiments, prepared figures and/or tables, authored or reviewed drafts of the article, and approved the final draft.

Lixin Sun performed the experiments, analyzed the data, prepared figures and/or tables, authored or reviewed drafts of the article, and approved the final draft.

Yun Zhao analyzed the data, prepared figures and/or tables, and approved the final draft.

Jingchun Yao analyzed the data, authored or reviewed drafts of the article, and approved the final draft.

Zhong Feng analyzed the data, prepared figures and/or tables, and approved the final draft.

Zhong Liu performed the experiments, authored or reviewed drafts of the article, and approved the final draft.

Guimin Zhang conceived and designed the experiments, performed the experiments, authored or reviewed drafts of the article, and approved the final draft.

Chenghong Sun conceived and designed the experiments, performed the experiments, prepared figures and/or tables, authored or reviewed drafts of the article, and approved the final draft.

The following information was supplied relating to ethical approvals (i.e., approving body and any reference numbers):

State Key Laboratory of Generic Manufacture Technology of Chinese Traditional Medicine provided full approval for this research.

The following information was supplied regarding data availability:

The raw measurements are available in the Supplementary Files.

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
