# Peer review of "UHPLC-ESI-QE-Orbitrap-MS based metabolomics reveals the antioxidant mechanism of icaritin on mice with cerebral ischemic reperfusion"

_PeerJ, doi:10.7717/peerj.14483_

## Round 0.1 · original submission · Minor Revisions

As you will see from their comments, both reviewers were positive. Reviewer-1 wishes you to address some concerns about the clarity of the figures and has some other points which should be easy to address.

Please address and state clearly in your revision the issue of infarct size and its potential impact on your data and interpretation. That way, the field can be the judge of this issue. However, it must be clearly signposted as a limitation of the work.

I think that reviewer-2 would prefer you to use a phrase such as 'protein levels' rather than expression.

Thanks for submitting this interesting study and i look forward to seeing the revisions.

·

Basic reporting

Studying the antioxidant effects of flavonoids is a classic pharmacological study, and in your publication on Chin Herb Med you have shown: ICT possessed significant neuroprotective effects in cerebral I/R mice, which might be related to prevent neuroinflammatory and oxidative. I personally think that this research is a good continuation and exploration. However, there are many obvious problems in the format and expression of the article, which require rigorous revision by researchers. The authors should be more cautious in their conclusions. The authors used whole brain homogenates to analyze metabolites, so the changes in metabolome products may be simply a result of the reduction in infarct size rather than a specific effect of ICT. Once again, because English is not my mother tongue, if these comments are difficult to understand, please forgive me.

Experimental design

Regarding the experimental design, there are mainly these questions: n=10 in this paper, wb n=10, which mice were used for the ratios of infarct volume (Fig. 1B), n=? In particular, many pictures do not clearly state the range of n, and it is not clear how you designed and distributed the experimental materials.

Validity of the findings

The data presented are relatively reliable, but conclusions should be drawn with greater caution. The cross-talk between signaling pathways and metabolic pathways is very complex, and it is difficult to judge the upstream and downstream relationship. The authors imply that ICT inhibits oxidative stress by changing metabolites, but the results in this paper do not support this view. Meanwhile, I suggest the author replace the WB representative band in Figure 6, especially Nrf2. Any conclusion should be based on the data, don't say more than what data could tell.

Additional comments

1. line 27 protective e?
2. If possible, it is recommended to add a conclusion or summary to the abstract.
3. Experimental materials and methods - It is recommended to add the number to the kits purchased by Nanjing Jiancheng, especially for a kit or material that has many methods for one test type, such as SOD, there are many types, and different test types will lead to different results.
4. FIG1 and FIG2 are not good -looking, you should find a way to solve the blank. Is Fig1, B, D Sham 0 reasonable? And it is recommended to Ba or Bb, or directly B, C. And the B it is recommended to add a ruler. Or you delete A directly and add CAS numbers to the material, which is actually more convenient and accurate.
5. Fig6. A KD and numbers are not uniform, and A contains multiple small pictures. The good choice is to include a, b. I don’t know why the name of AB's horizontal coordinate group is different. One is normal and the other is called Sham.
6. There is a significant gap between multiple metabolites in each group. Which is the main? Is the oxidation stress that causes changes in metabolites, or is the oxidation stress caused by changes in metabolites? Or what do you think is the target of ICT?
7. Personally, the conclusions are too absolute for the results of unwritten components of metabolism. And because this study does not involve inhibitors or siRNA, the shortcomings of research should be added more descriptions.

·

Basic reporting

The work presented is interesting, it brings innovations to the area, it has a good quality of English, the figures and tables are in excellent quality and they present the data in a consistent way. Conclusions and discussion are adequate and references are current and well used throughout the work.The working hypothesis was well presented and the experiments were adequate to answer the proposed question.

Experimental design

The question proposed and discussed in the paper is original, the data obtained are of great relevance and contribute to the development of the proposed theme.The experiments presented were carried out and discussed with scientific rigor, as well as their statistical analysis. My only reservation is due to the presentation of the immunocontent data of NRF2, KEP-1, HO-1 and NQO-1 where the author talks about expression and the correct thing would be to talk about immunocontent

Validity of the findings

I believe that the data presented are robust and that they were well used in the discussion and confirmation of the hypothesis.The conclusions are adequate and do not extrapolate what was observed in the results and present in the literature.

Additional comments

A very interesting, current and relevant work for the area.

---

## Round 0.2 · accepted · Accept

Thanks for addressing these issues.